

# Age-based and reproductive biology of the Pacific Longnose Parrotfish *Hipposcarus longiceps* from Guam

Brett M. Taylor[1] and Eric Cruz[2]

[1] Joint Institute for Marine and Atmospheric Research, University of Hawaii, Honolulu, HI, United States of America

[2] Guam Field Office, NOAA Pacific Islands Fisheries Science Center, Barrigada, Guam, United States of America

## ABSTRACT

The Pacific longnose parrotfish *Hipposcarus longiceps* (Valenciennes 1840) represents a prime fishery resource throughout much of the tropical Pacific. In this study, we sampled the species from the Guam commercial fishery market across five consecutive years to characterize reproductive and age-based demographic information imperative for informed fishery management. Compared with other parrotfishes, this species was found to be large-bodied, but has only a moderate life span of 10+ years. *Hipposcarus longiceps* was confirmed as a diandric protogynous hermaphrodite with highly sex-specific growth patterns and an overall mean asymptotic length of 434 mm fork length (FL). Females were estimated to reach median maturity at 329 mm FL (2.4 years) and have a median length at female-to-male sex change of 401 mm FL. Life-history trait values derived here were used to update previous models relating life history and vulnerability to overexploitation. We found that enhancement of just one species' trait values improved model fits considerably, which strengthens the conclusion that life-history traits are a strong determinant of species' vulnerability in the parrotfishes. This information is an imperative complement to other data sources facilitating formal stock assessment of a key fishery target.

Corresponding author
Brett M. Taylor,
brett.taylor@noaa.gov

## INTRODUCTION

Parrotfishes (family Labridae; tribe Scarinae) are among the most common groups of taxa represented in coastal fisheries harvests of the tropical Indo-Pacific region (*Rhodes, Tupper & Wichilmel, 2008*; *Taylor & Choat, 2014*). At the US Pacific Island Territory of Guam, parrotfishes (known collectively in native Chamorro as *palaksi* or *laggua* based on body phase coloration; *Kerr, 1990*) have been important fishery resources in both historical and contemporary times (*Houk et al., 2012*). Overall, parrotfishes represent approximately 20% of the total Guam nearshore commercial harvest by weight, making them the second most common taxonomic group (NOAA Commercial Fisheries BioSampling Program, 2009–2017, unpublished data). Since the 1980s, nighttime spearfishing with the aid of SCUBA has been the most prevalent means of harvesting reef-associated species commercially,

with parrotfishes being particularly susceptible to this method because of their behavior of sleeping in easily located habitats on coral reefs.

Among the 20+ parrotfish species that occur at Guam, the most heavily targeted species by weight is *Hipposcarus longiceps* (native Chamorro name *gualafi*). This species is also the third most targeted species by weight in the SCUBA-spear fishery, behind the unicornfishes *Naso unicornis* (Forsskål 1775) and *N. lituratus* (Forster 1801). *Hipposcarus longiceps* is phylogenetically unique (only two extant *Hipposcarus* exist; *Choat et al., 2012*), forms large schools for both foraging and reproduction (*Colin, 2012*), and is one of the largest species of parrotfish that occurs at Guam, sometimes reaching nearly 60 cm fork length. *H. longiceps* is considered a scraping detritivore and microphage (*Bellwood & Choat, 1990*; *Choat, Robbins & Clements, 2004*), and is most commonly found in schools roving back and forth across the fore reef environments adjacent to reef flats or lagoons on Guam (*Taylor et al., 2014*). The species can be found sleeping in schools, sporadically spaced across the reef within a relatively small area (B Taylor, pers. obs., 2011, 2013), and are thus behaviorally predisposed to be prime targets to nighttime spear fishers. As with many scarine labrids, *H. longiceps* adults have two distinct, albeit subtly different, color phases (initial and terminal) reflecting female and male sexes (see *Colin, 2012*), except that primary males (individuals that originally developed as males) are found in both color phases. Terminal phase primary males have undergone metamorphosis of body coloration during their ontogeny without changing sex, as can be confirmed histologically (*Taylor & Choat, 2014*).

Life-history traits are known to strongly influence species' responses to exploitation (*Adams, 1980*; *Abesamis et al., 2014*). This relationship between vulnerability to overexploitation and life-history trait values has been well-documented in the parrotfishes, specifically across reefs of Guam (*Taylor et al., 2014*), whereby age at female maturation was the strongest predictor of spatial and temporal patterns of fishery-induced demographic changes across species. In that analysis, *H. longiceps* was found to be among the most vulnerable species with corresponding life-history traits (e.g., length and age at maturity, length at sex change, maximum length, growth rate, maximum age) generally achieving higher values compared with other species. However, that study incorporated life-history data from Pohnpei (Federated States of Micronesia) as a proxy for Guam *H. longiceps*. Life-history traits can vary considerably and predictably within regions, whereby we would expect Guam populations to be larger-bodied and potentially longer-lived based on island geomorphology and latitude (*Taylor, 2014*).

The purpose of this study is to derive detailed age-based and reproductive demographic information for the most commonly targeted parrotfish species in Guam's commercial fishery. This study provides initial insight to stock status, facilitates future formal stock assessment, and updates models of vulnerability to better reflect locally-specific parameter values. Market-based surveys and collections spanning nearly six years were undertaken to derive parameters of growth, life span, reproductive ontogeny, mortality, and spawning periodicity.

## MATERIALS & METHODS

### Study area and sampling protocol

The major commercial vendor of reef-associated fisheries was surveyed on the US Pacific Island Territory of Guam (13°28′N, 144°45′E), from Aug 2010 to May 2016, through the NOAA Commercial Fisheries BioSampling Program (CFBP). All research was carried out under permit UOG1202 issued by the University of Guam Institutional Animal Care and Use Committee. Survey methods and sampling protocols are detailed in *Sundberg et al. (2015)*. An average of five specimens of *Hipposcarus longiceps* was purchased monthly for biological analysis. Specimens were processed immediately after purchasing. For each specimen, general location of capture and fishing method were obtained from the vendor; measurements of fork length (FL, nearest mm) and total body weight (TW, g) were also obtained. Sagittal otoliths and gonad lobes were surgically removed from each specimen for subsequent processing. Otoliths were cleaned in ethanol and stored dry in plastic vials. Gonad lobes were weighed to the nearest 0.001 g (GW). Entire gonads or sections (1 cm thick) of gonad material from the mid-sections of gonad lobes were removed and stored using individually-labelled histological cassettes in a buffered 10% formalin solution or Shandon® Glyofix® (Thermo Fisher, Waltham, MA, USA).

### Age determination and growth

One sagittal otolith from each specimen's pair was randomly selected for age analysis. Each otolith was affixed to a glass slide using thermoplastic glue (Crystalbond 509®; Aremco Products, Inc., Valley College, NY, USA) so that the primordium was focused just inside the slide edge, and the direction of the sulcus ridge was perpendicular to the slide edge. The otolith was ground to the primordium using a 600 grit diamond lap on a grinding wheel along the longitudinal axis to the edge of the slide. The otolith was then removed using a hotplate and reaffixed with the newly flat surface down and ground to produce a thin (∼250 µm) transverse section encompassing the core material. Annuli (alternating translucent and opaque bands) along the face of the transverse sections were counted using transmitted light on a stereo microscope on three separate occasions. Annual ages were assigned when at least two counts were in agreement, which occurred for all but four specimens. Counts of those four specimens differed by only 1 annuli (e.g., 9, 11, 10), so the middle value (e.g., 10) was assigned.

The assumption of annual deposition of increments in transverse sections of *H. longiceps* otoliths was tested using edge-type analysis. The optical characteristic of the otolith margin was scored as opaque (slower growth, denser material) or translucent (faster growth, less-dense material). Decimal ages (fish ages to a finer resolution than one year) were not derived because the species spawns and presumably recruits throughout the year; hence, there is no appropriate mean birth date for specimens.

Sex-specific and combined patterns of growth were examined using length-at-age data fitted with the von Bertalanffy growth function (VBGF), represented by:

$$L_t = L_\infty[1 - e^{-k(t-t_0)}]$$
where $L_t$ is the predicted mean FL (mm) at age $t$ (years), $L_\infty$ is the asymptotic FL, $k$ is the coefficient used to describe the curvature of fish growth towards $L_\infty$, and $t_0$ is the hypothetical age at which FL is equal to zero, as described by $k$. The growth model was constrained to a common FL at settlement of 15 mm (following *Bellwood & Choat, 1989*) to account for early growth of juveniles since specimens below 188 mm FL were not obtained. Male and female growth patterns were compared by plotting 95% bivariate confidence ellipses surrounding paired estimates of $L_\infty$ and $k$ following *Kimura (1980)*.

## Mortality

Age composition and instantaneous rates of mortality were explored through age frequency distributions. The sex-specific frequency of individuals sampled from the fishery was plotted by length and age classes. The instantaneous rate of total mortality ($Z$, yr $^{-1}$) was estimated two ways. First, we used a Gaussian likelihood-based linear catch curve with knife-edge selectivity. This method estimated $Z$ as the absolute value of the slope of a line fitted to the natural logarithm of observed frequency of harvested individuals at age $t$ ($O_t$) for corresponding age classes above $t_c$, the age at which fish are assumed fully recruited into the fishery (*Beverton & Holt, 1957*). We conservatively defined $t_c$ as one plus the age at peak observed frequency.

We then used a multinomial likelihood-based catch curve using logistic selectivity by minimizing the multinomial negative log-likelihood $\lambda$ associated with $O_t$ and the expected proportion ($P_t$) of fish at or above $t_c$

$$\lambda = -\sum_{t=1}^{A} O_t \ln(P_t)$$

$$P_t = \frac{C_t}{\sum_{t=1}^{A} C_t}$$

where $A$ refers to the maximum age assumed in the analysis, arbitrarily set much larger than the observed maximum age, thereby avoiding calculation of a "plus group." $C_t$ is the expected catch at age $t$, calculated using the Baranov catch equation:

$$C_t = \frac{F_t}{Z_t} N_t (1 - e^{-Z_t})$$

$$Z_t = F_t + M$$

$$F_t = FV_t$$

$$V_t = \frac{1}{1 + e^{\frac{-\ln(19)(t-A_{50})}{(A_{95}-A_{50})}}}$$

where $F_t$ is the fishing mortality at age $t$, $F$ is the "full" instantaneous fishing mortality, and $V_t$ is the estimated gear selectivity at age $t$ following a logistic function with selectivity parameters $A_{50}$ and $A_{95}$, the ages at 50% and 95% selectivity, respectively. The equation of *Hoenig (1983)* was used to estimate natural mortality ($M$) based on the maximum observed age ($t_{max}$), with the assumption that $t_{max}$ reflects the true life span. Thus, $Z = M$ in the equation:

$$Z = e^{(1.46 - 1.01 * \ln(t_{max}))}.$$

## Reproduction

Sampled gonads were histologically processed at the John A. Burns School of Medicine at the University of Hawaii. Fixed sections of gonad tissue were imbedded with paraffin wax, sectioned transversely at 6 $\mu$m, and stained on microscope slides with haematoxylin and eosin following standard procedures. Samples were viewed under compound and stereo microscopes to identify features determining sex and maturation stage following criteria detailed in *West (1990)* and using standardized terminology from *Brown-Peterson et al. (2011)*. Particular attention was given to (1) presence or absence of post-ovulatory follicles (POFs) as confirmation of prior spawning, (2) proof of post-maturational sex change in the form of proliferating spermatogenic material in the presence of degenerative vitellogenic oocytes, and (3) characteristics of an ovarian lumen in male testes signifying secondary (prior sex change from female to male) versus primary (individual originally developed as male) male development.

Length (mm, FL) and age (years) at 50% female maturation and sex change were estimated by fitting a two-parameter logistic curve to the proportion of mature individuals per length and age class (20 mm and 1-year bins). The logistic curve was as follows: $P_L = [1 + e^{-\ln(19)(L - L_{50})/(L_{95} - L_{50})}]^{-1}$, where $P_L$ ($P_t$ for age at maturity) represents the estimated proportion of mature females at a given length ($L$), and $L_{50}$ and $L_{95}$ ($t_{50}$ and $t_{95}$ for age at maturity) represent the FL when 50% and 95% of the population is reproductively mature. The same procedure was carried out to estimate length at sex change, where lengths at 50% and 95% sex change are denoted by $L_{\Delta 50}$ and $L_{\Delta 95}$. Corresponding 95% confidence intervals (CI) for each parameter were derived by bootstrap resampling using 1,000 iterations.

A gonadosomatic index (GSI) was calculated as the ratio of gonad-to-gonad-free body weight for each mature female in an active (not resting or spent) phase. We then examined temporal trends in GSI across calendar months and lunar days by plotting raw GSI values pooled across these timeframes. Occurrence of POFs or hydrated oocytes was also examined across these timeframes to indicate timing and frequency of spawning activity at two temporal scales.

## Re-examination of vulnerability to overexploitation

The influence of life-history traits on the incidence and magnitude of vulnerability to overexploitation was examined for Guam parrotfishes in *Taylor et al. (2014)*, using patterns of abundance and mean length across spatial and temporal gradients of fishing pressure. *H. longiceps* was found to be among the more vulnerable species in the Guam assemblage, but life-history trait values assessed in that study were derived from a surrogate location of Pohnpei, Micronesia. Patterns of life-history variation associated with geomorphological differences between Guam and Pohnpei suggest that Guam *H. longiceps* would be larger and longer-lived (*Taylor, 2014*). Hence, we repeated the prior analyses of incidence and magnitude of vulnerability by replacing the Pohnpei *H. longiceps* data with newly derived Guam data from the present study. Specifically, the traits age at maturity ($t_{50}$), length at female maturity ($L_{50}$), length at sex change ($L_{\Delta 50}$), mean maximum length and age (mean of the largest [$L_{max}$] and oldest [$t_{max}$] quartile of the sampled population), and

modeled growth (in mm) from age 1 to age 3 ($L_{1-3}$) were assessed for their power to predict vulnerability to overexploitation across species (details of analysis in *Taylor et al., 2014*). We hypothesized that model fits would improve with the use of regional-specific Guam data in lieu of population parameters from Pohnpei.

## RESULTS

### Age determination and growth

From Dec. 2009 to May 2016, a total of 3,369 individual *H. longiceps* were surveyed from the Guam commercial fishery. Ninety-seven percent of these were harvested by spearfishing with SCUBA, 1.6% by spearfishing with snorkel, and 1.4% by gill net or unknown method(s). From these, 330 were purchased and sampled for life-history analysis. The smallest and largest specimens sampled were 188 mm (130 g) and 514 mm FL (2,650 g), respectively, and the average length and weight of sampled fish was 366 mm FL ($\pm 75$ SD) and 1,073 g ($\pm 593$ SD). The relationship between FL (mm) and TW (g) was described by $TW = 1.739 \times 10^{-5}(FL)^{3.018}$ (earlier derivation of this relationship for cm presented in *Kamikawa et al., 2015*).

Transverse sections of sagittal otoliths revealed overall banding patterns that were highly consistent in deposition among specimens (Fig. 1). These bands were confirmed to be annular by edge-type analysis, whereby opaque zones, representing slow growth, were deposited in the winter months (peaked in December) and were non-existent at the otolith edge from April to July (Fig. 1). Spacing of the first increment was variable among specimens because individuals likely recruited throughout the year (see 'Reproduction' section). Because of this, individual ages throughout are reported in years as integers, representing the total number of annular bands.

Sampled female *H. longiceps* ranged from 188 to 470 mm FL (mean = 345 mm $\pm$ 57 SD), and sampled males ranged from 242 to 514 mm FL (mean = 416 mm $\pm$ 59 SD; Fig. 2A). Sex-specific length ranges overlapped considerably but frequency distributions demonstrated that males were, on average, much larger than females (two sample $t$-test, df = 300, $t = -10.66$, $P < 0.001$). Primary males (based on histological features of testis; represents both IP and TP primary males) spanned the full size range of all males. The modal age for females was two years, and proportional frequencies dropped drastically beyond that age group (mean age = 2.8 years). For males, the modal age ranged from 2–5 years for which there was approximately equal frequencies (mean age = 4.0 years). Males were generally older than females, and the oldest individuals collected for each sex were ten years for males and nine years for females (Fig. 2B). Seven of the aged specimens were considered sex unknown, for which macroscopic designations were undetermined, and no histology was performed.

The overall modal age was two years for all specimens combined. The instantaneous rate of total mortality ($Z$) based on the linear age-based catch curve with knife-edge selectivity was 0.476 yr$^{-1}$ (95% CI [0.375–0.577]), and the $Z$ estimate from the multinomial catch curve was 0.402 yr$^{-1}$ (95% CI [0.287–0.518]) (Fig. 2B). *Hoenig*'s (*1983*) estimate of $M$ was 0.421 yr$^{-1}$, presuming the maximum age of 10 years represents the natural maximum
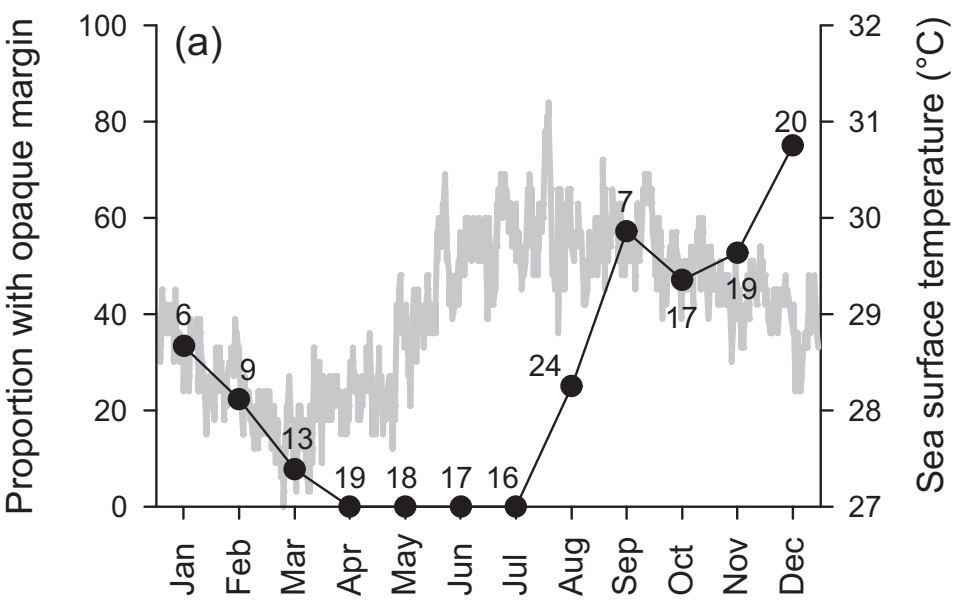

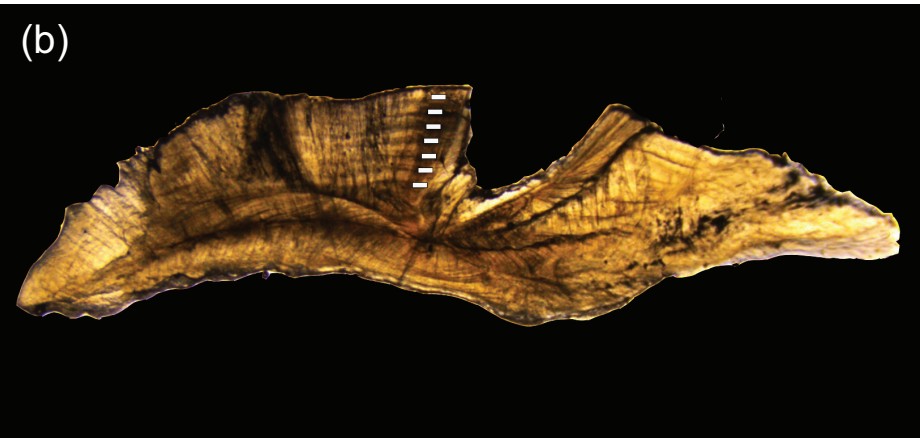

**Figure 1** (A) Frequency of opaque edge deposition by month in transverse sections of *Hipposcarus longiceps* otoliths plotted with sea surface temperature (satellite derived, Terra MODIS database, 2010). Numbers associated with data points indicate sample sizes. (B) Photomicrograph of a transverse section of a *Hipposcarus longiceps* otolith viewed with transmitted light. Opaque annuli are denoted by rectangles.

age in the unexploited wild population. This value was greater than the multinomial catch curve estimate of $Z$, suggesting that estimates of $Z$ and $M$ from the fishery-dependent samples are not robust.

Overall, *H. longiceps* grew rapidly in the first year, achieving over 300 mm in many specimens with one annuli (Fig. 3). The general asymptotic length was reached in year four, whereas growth profiles separated for the males and females at year three. Males reached a mean asymptotic length of 466 mm FL, whereas females reached 396 mm FL. Sex-specific growth profiles differed considerably, as shown by non-overlapping bivariate

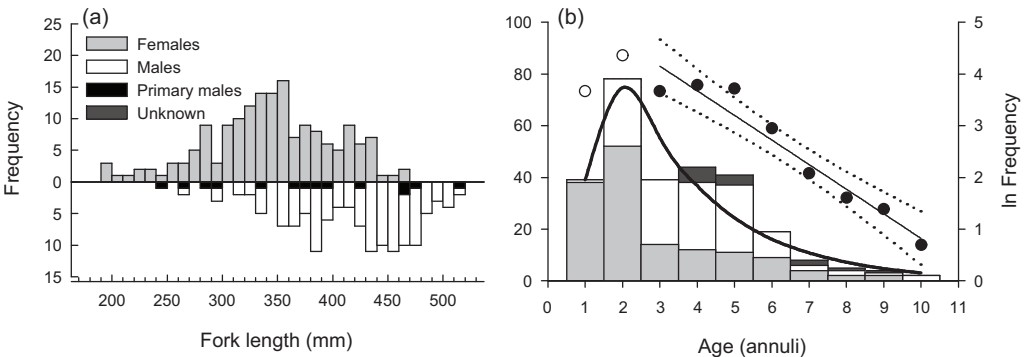

**Figure 2** **(A) Sex-specific length- and (B) age-frequency distributions of sampled *Hipposcarus longiceps* from Guam.** Data points and linear regression line (with 95% confidence limits) in (B) represent the age-based catch curve ($y = mx + b$) derived from natural log-transformed catch at age values. Open circles indicate age classes considered not fully recruited to the fishery. The curved line represents the expected proportion of individuals at age from the multinomial catch curve with logistic selectivity.

confidence ellipses surrounding estimates of $L_\infty$ and $K$ (Fig. 3). The overall growth parameters (sexes combined) for *H. longiceps* in Guam were $L_\infty = 434$ mm FL, $K = 0.786$ yr$^{-1}$, and $t_0 = -0.045$ years (Table 1).

## Reproduction

A total of 270 specimen gonads were processed histologically for assignment of sex and reproductive stages. These individuals ranged from 195 to 514 mm FL and included 150 females and 120 males. Three of the males were considered to be in the process of sexual transition from female to male and displayed degenerative oocytes in the presence of proliferating spermatogenic material (Fig. 4). Macroscopic field designations of sex were accurate 80% of the time. The length and age at 50% female maturation was found to be 329 mm FL (95% CI [313–345]) and 2.4 years (95% CI [2.2–2.7]), respectively (Figs. 5A, 5B). Length at sex change was determined to be 401 mm FL (95% CI [381–420]; Fig. 5C). Age at sex change was not calculable because sex change is a highly length-based, socially-mediated process, whereby many females do not undergo sex change throughout extended life spans.

We found few insights to spawning periodicity or frequency across calendar months or lunar phases. Mature, active females had GSI values up to 4% of their gonad-free body weight (Fig. 6). We found no annual or lunar trend in GSI values, and we found a high proportion of mature, active females containing post ovulatory follicles and/or hydrated oocytes in every calendar month and across each lunar phase (Fig. 6). The proportional presence of POFs and hydrated oocytes appears to correlate with the maximum observed GSI across both annual and lunar timescales.

## Vulnerability analyses

Trait values for Guam *H. longiceps* were greater than those from Pohnpei for five of the six life-history traits examined in vulnerability models, based on *Taylor et al. (2014)*. As predicted, a majority (ten out of twelve) of models examining the predictive power
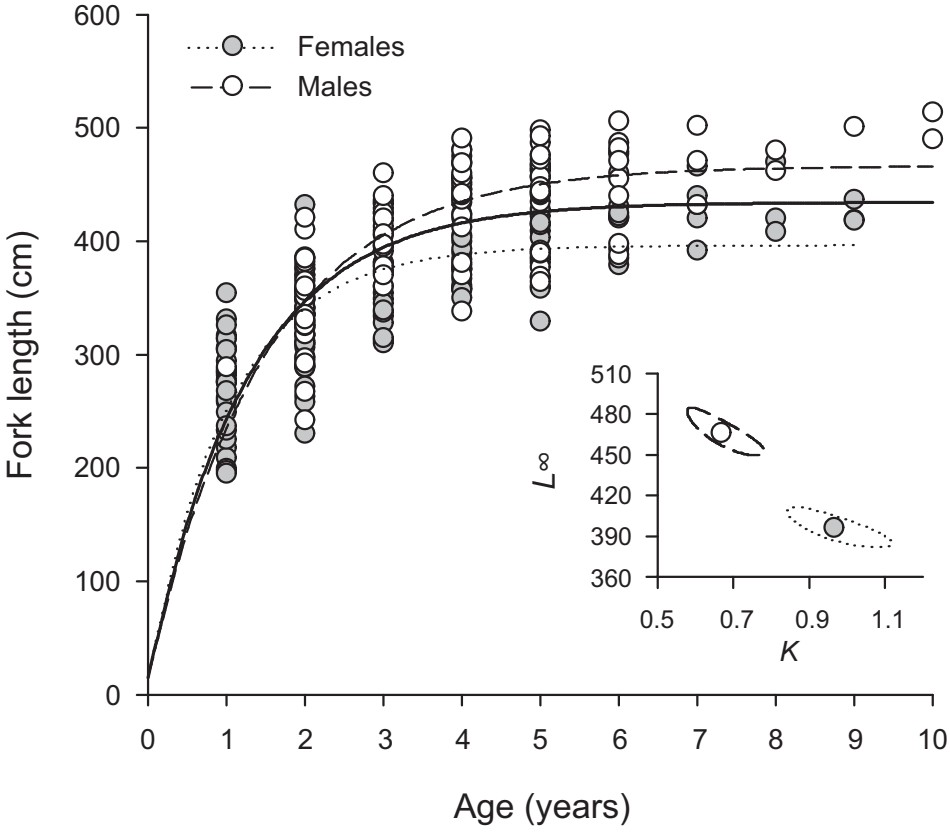

**Figure 3** **Sex-specific patterns of growth, modelled using the von Bertalanffy growth function, for** *Hipposcarus longiceps* **from Guam (** $n = 265$ **).** $L_\infty$, asymptotic fork length; $K$, coefficient used to describe the curvature of fish growth towards $L_\infty$. Inset, ellipses represent the 95% confidence intervals surrounding estimates of $K$ and $L_\infty$ for each sex. Points within confidence ellipses represent mean estimated values.

of life-history traits on vulnerability to overexploitation had an improved fit with the substitution of Guam *H. longiceps* data (Supplemental Information). Only the predictive power of age at maturity ($t_{50}$) decreased, although this trait remained the most influential for explaining incidence of response. Overall, the average variance explained increased by 4% (6% without $t_{50}$) for models predicting incidence of response and by 6.5% (11% without $t_{50}$) for models predicting magnitude of response (Supplemental Information).

## DISCUSSION

This study provides the most comprehensive assessment of age-based and reproductive biology for *Hipposcarus longiceps*, a species that is commonly harvested in reef-associated subsistence and commercial fisheries throughout the tropical Pacific. In brief, *H. longiceps* is a comparatively large-bodied parrotfish with a moderate life span and fairly rapid initial growth to targeted size classes. We found no evidence of distinct reproductive periodicity at any temporal scale, but did identify several features of reproductive demography considered

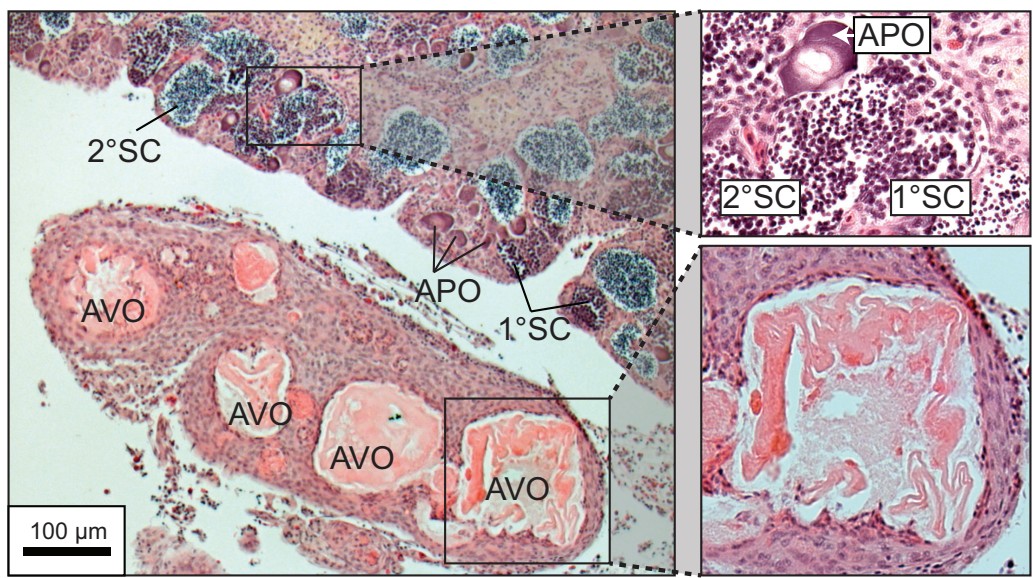

**Figure 4** Photomicrograph of a histological section of a 370 mm specimen undergoing sexual transition from a functionally mature female to a maturing male, demonstrating structural characteristics of degenerative vitellogenic and primary oocytes (AVO, atretic vitellogenic oocyte; APO, atretic primary oocyte) in the presence of proliferating spermatogenic material (1° SC, primary spermatocytes; 2° SC, secondary spermatocytes).

**Table 1** Summary of life-history trait estimates for *Hipposcarus longiceps* from Guam. Associated 95% confidence intervals are presented in parentheses where appropriate.

| Trait | Females | Males | Combined |
|---|---|---|---|
| $L_\infty$ (mm) | 396 (382–411) | 466 (450–485) | 434 (422–446) |
| $K$ (yr$^{-1}$) | 0.97 (0.84–1.12) | 0.67 (0.58–0.78) | 0.79 (0.71–0.87) |
| $t_0$ (yr) | −0.04 | −0.05 | −0.04 |
| $n$ | 144 | 121 | 279 |
| $L_{50}$ (mm) | 329 (313–345) | – | – |
| $t_{50}$ (yr) | 2.4 (2.2–2.7) | – | – |
| $L_{\Delta 50}$ (mm) | – | – | 401 (381–420) |
| $Z$ (yr$^{-1}$) linear | – | – | 0.476 (0.375–0.577) |
| $Z$ (yr$^{-1}$) multinomial | – | – | 0.402 (0.377–0.489) |

**Notes.**
$L_\infty$, asymptotic length; $K$, growth coefficient; $t_0$, hypothetical age when length equals zero; $n$, sample size; $L_{50}$, length at 50% sexual maturity; $t_{50}$, age at 50% sexual maturity; $L_{\Delta 50}$, length at 50% sex change; $Z$ linear, instantaneous total mortality rate estimated from the linear age-based catch curve; $Z$ multinomial, instantaneous total mortality rate estimated from the logistic multinomial catch curve.

unique among the parrotfishes. Below, we discuss our findings in a phylogenetic, behavioral, and fisheries context.

Our results provide the first confirmatory evidence of functional diandric protogynous sex change in the genus *Hipposcarus*, based on histological examination of gonad structures and sex-specific size and age distributions. This result is not surprising, given that protogynous hermaphroditism dominates among the parrotfishes. However, *H. longiceps* is
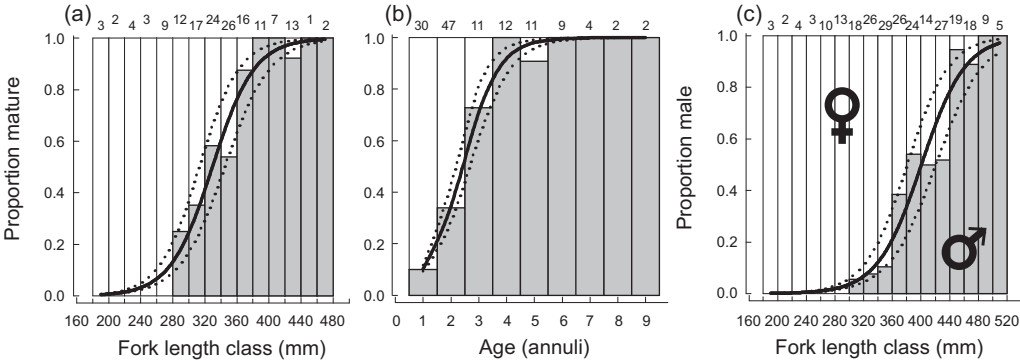

**Figure 5** Plots of (A) fork length and (B) age at 50% sexual maturity, and (C) fork length at 50% sex change, along with associated logistic ogives describing the patterns of maturation and sex change for female *Hipposcarus longiceps* from Guam. Numbers above sampling bins represent respective sample sizes. Solid lines represent the best-fit models, dotted lines represent the associated 95% confidence intervals.

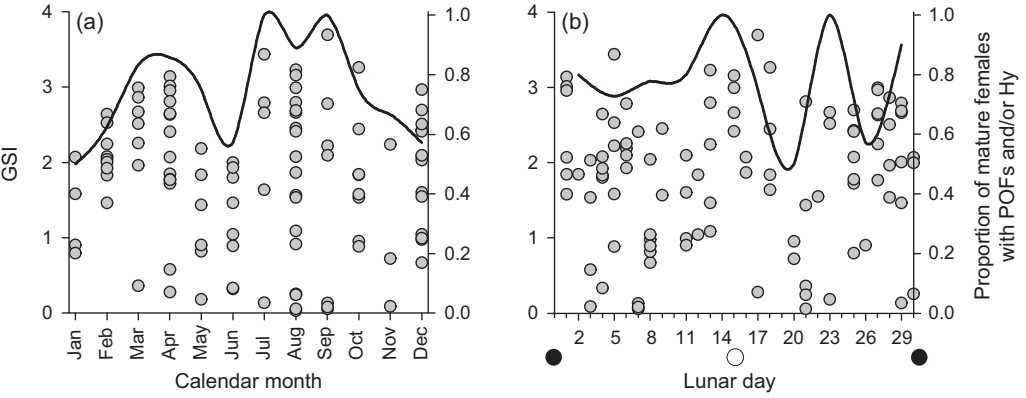

**Figure 6** Plots of raw gonadosomatic index values (left *y*-axis) for mature, active female *Hipposcarus longiceps* (*n* = 103) across (A) calendar months and (B) lunar days. Solid lines represent the proportion of mature females with the presence of post ovulatory follicles (POFs) and/or hydrated oocytes (right *y*-axis), indicating the recent or imminent occurrence of spawning.

phylogenetically distinct, and confusion over reproductive function in scarine labrids is not uncommon (*Sadovy de Mitcheson & Liu, 2008*); examples of functional gonochorism exist (*Robertson, Reinboth & Bruce, 1982*; *Hamilton, Adams & Choat, 2008*). Another peculiar feature of the reproductive demography of *H. longiceps* is the variable rate of primary males with initial phase coloration across populations. Male specimens from the Pohnpei sample reported in *Taylor & Choat (2014)* contained 61% initial phase primary males, whereas only 16% of the specimens from Guam (present study) were initial phase primary males. The presence and proportion of initial phase primary males has a strong social basis and a primary male strategy becomes more prevalent with increased social group size (*Munday, White & Warner, 2006*). Observed differences between populations may reflect lower

densities in Guam compared with Pohnpei, although this cannot be resolved at present. However, this has been observed for subpopulations on the Great Barrier Reef, where reef structures with high densities of the bullethead parrotfish *Chlorurus spilurus* (Valenciennes 1840) (formerly *Chlorurus sordidus* (Forsskål 1775)) contained greater proportions of primary males (*Gust, 2004*).

Despite sampling across all calendar months over a 5+ year period, our data resolution was not great enough to derive insight to spawning periodicity for *H. longiceps*, either on an annual or a lunar time scale, based on GSI patterns. However, there was a very high proportion of mature females with post-ovulatory follicles and/or hydrated oocytes across all months (mean = 75%) and lunar periods (mean = 78%), suggesting that spawning activity occurred frequently throughout each year. Detailed descriptions of spawning behavior have been documented by *Colin & Bell (1991)* from Enewetak Atoll in the Marshall Islands and by *Colin (2012)* from Palau. In both accounts, groups were observed spawning at seaward facing reefs after high tide, often following the full or new moon lunar phases. Males appear to patrol territories and engage in courtship behavior. Both of these locations where spawning has been observed are geomorphologically different from Guam, and reef morphology can strongly influence reproductive behavior in labrids (*Warner & Hoffman, 1980*). However, the level to which the reproductive ecology of *H. longiceps* in Guam may reflect or differ from that previously reported from elsewhere remains unclear.

The present study also provides the first validation of annual periodicity in otolith increments of *Hipposcarus* using edge-type analysis. Maximum age from 279 specimens was found to be 10 years on Guam. Other maximum age estimates from *H. longiceps* are 6 years from Pohnpei (*Taylor & Choat, 2014*) based on 65 specimens, 12 years from the northern Great Barrier Reef (*Choat & Robertson, 2002*) based on 23 specimens, and 5 years from Solomon Islands (*Sabetian, 2010*) based on 67 specimens. It is now clear that *H. longiceps*, while among the larger parrotfishes, has only a moderate life span potentially stretching into the low teens and naturally varying across environments, latitude, and different fishing pressures. Both males and females were well-represented in the older age classes, suggesting that sex change is a largely length-based, socially-mediated process and many older females never undergo sex change through their life spans.

We found poor compatibility among mortality estimates in this study. The estimate of natural mortality from the *Hoenig (1983)* equation exceeded the estimate of total mortality from the more robust multinomial catch curve, implying that the species on Guam faces no fishing pressure. This does not conform with two facts: (1) *H. longiceps* is among the most prevalently harvested species in the commercial fishery, and (2) mean harvested length has declined steadily over at least a 30-year period (*Lindfield, McIlwain & Harvey, 2014*). We suspect discrepancies arise for two reasons. First, the dynamics of the fishery may make estimating total mortality from fishery-dependent data challenging. The night-time harvest of *H. longiceps* is highly seasonal. The period of greatest harvest coincides with the calmest ocean conditions, approximately from May through August, where average monthly harvest weight more than doubles. Further, the inherent nature of the spear-fishery is highly selective, and larger individuals are preferred based on market value. Sampling was

also not conducted over a discrete time period, allowing potential fluctuations in annual recruitment to influence catch curve estimates. A fishery-independent sampling strategy may rectify some of these issues and improve resolution for instantaneous mortality rates. Secondly, our sampling program possibly did not capture true maximum age, thereby increasing our estimate of natural mortality. The present sample size relative to number of age classes suggests that the level of precision for estimating maximum age was appropriate (*Kritzer, Davies & Mapstone, 2001*), but truncation of age structure cannot be discounted when sampling from historically harvested stocks. Ultimately, estimates of total mortality derived in this study should be interpreted with caution.

Replacing data from Pohnpei with data from Guam in an otherwise Guam-specific analysis of vulnerability, led to considerable improvements in fit for models linking life-history traits with vulnerability to overexploitation. This strengthens the conclusion from *Taylor et al. (2014)* that life-history traits are strong predictors of vulnerability to overexploitation in parrotfishes. However, it also highlights the danger in, and fisheries implications of, borrowing life-history data from other regions. Here, trait values between Pohnpei and Guam populations did not differ greatly, but the direction of differences was generally consistent across traits. This directionality was reflected in improved model fits because *H. longiceps* was among the most vulnerable species, and trait values from Guam were, as predicted, generally greater than those from Pohnpei. Contemporary stock assessment strategies for tropical fisheries rely on quality biological information and region-specific information can remove much of the guess-work involved in data-poor fisheries management (*Newman et al., 2016*; *Prince et al., 2015*).

## CONCLUSIONS

*Hipposcarus longiceps* is a priority species in the Guam reef-associated commercial fishery. By deriving detailed age-based and reproductive information for the exploited stock from Guam, this study lays a biological framework for understanding population dynamics in the species and is directly applicable for fishery managers.

## ACKNOWLEDGEMENTS

The authors thank the Guam Biosampling Team for their immense efforts and TT Jones and D Kobayashi for comments on an earlier draft of the manuscript. Formal reviews by Jason Morton and one anonymous reviewer improved the manuscript.

### Funding

This study was funded by the National Marine Fisheries Service Biosampling Initiative and the Joint Institute for Marine and Atmospheric Research Territorial Biosampling Project 6105137. The funders had no role in study design, data collection and analysis, decision to publish, or preparation of the manuscript.

## Grant Disclosures

The following grant information was disclosed by the authors:

National Marine Fisheries Service Biosampling Initiative.

Joint Institute for Marine and Atmospheric Research Territorial Biosampling Project: 6105137.

## Competing Interests

The authors declare there are no competing interests.

## Author Contributions

- Brett M. Taylor conceived and designed the experiments, performed the experiments, analyzed the data, contributed reagents/materials/analysis tools, wrote the paper, prepared figures and/or tables, reviewed drafts of the paper.
- Eric Cruz conceived and designed the experiments, performed the experiments, contributed reagents/materials/analysis tools, reviewed drafts of the paper.

## Animal Ethics

The following information was supplied relating to ethical approvals (i.e., approving body and any reference numbers):

All research was carried out under permit UOG1202 issued by the University of Guam Institutional Animal Care and Use Committee.

## Data Availability

All raw data used for figures and analyses in the manuscript is supplied as Supplementary Information. All data used in this publication are from Pacific Islands Fisheries Science Center, 2017: Guam Commercial Fisheries BioSampling (CFBS), https://inport.nmfs.noaa.gov/inport/item/5625.

## Supplemental Information

Supplemental information for this article can be found online at http://dx.doi.org/10.7717/peerj.4079#supplemental-information.

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
