# Peer review of "Age-based and reproductive biology of the Pacific Longnose Parrotfish Hipposcarus longiceps from Guam"

_PeerJ, doi:10.7717/peerj.4079_

## Round 0.1 · original submission · Minor Revisions

I have heard back from two reviewers, both of whom offered many constructive comments. Please consider these carefully (and don't forget the annotated PDF file from reviewer 2), and I look forward to seeing your revised version.

·

Basic reporting

No comment

Experimental design

No comment

Validity of the findings

No comment

Additional comments

The paper ‘Age-based and reproductive biology of the Pacific longnose parrotfish Hipposcarus longiceps from Guam’ by Taylor and Cruz provides important demographic information about a commercially important species. The manuscript is written very well. The Introduction provides an excellent coverage of the background information and sets up the reasons for this study in the context of information gaps. The Methods and Results were explained well and, despite being unfamiliar with a couple of the analyses performed, literature support was provided to justify their use. I particularly liked the clear structure of the Discussion, which clearly outlined the importance of the findings in the context of relevant literature. Thank you to the authors for providing a manuscript that was enjoyable to read.

I have very few suggestions for improving this manuscript. However, the comments listed below should be considered…

Abstract:
(i) Line 20 – “… and tend to change sex to males…” This is rather ambiguous. I suggest rewording this.

Introduction:
(i) Lines 43-44 (and elsewhere) – The common name for this species is stated in the title so I do not see a reason for it to be included frequently throughout the manuscript. All references to the common name should be changed to the scientific name. This is the second time the common name for this species is mentioned in this paragraph.
(ii) Lines 45-46 – What is meant by “… among the larger species that occurs in Guam…”? Larger species of what? Of parrotfishes?
(iii) Lines 55-56 – “… as can be confirmed histologically.” This needs to be referenced.
(iv) Lines 60-61 – What is meant by “…whereby age at female maturation had the strongest relationship to…”? Does this mean they mature earlier?
(v) Lines 62-63 – Similar to my comment above, what does “… with corresponding life-history traits generally at the far-end of the spectrum.” actually mean? Does this mean at the lower/higher ends of the spectrum or at both ends? Also, what life-history traits are actually getting referred to here? Is it age/size at maturation and sex change?
(vi) Lines 66-67 – The way this sentence is constructed it implies that Taylor (2014) suggests that the Guam populations will be larger-bodied and longer-lived. Is this actually the case? If so, it may be more appropriate to place this reference at the end of the sentence. If not, then an additional reference is needed at the end of the paragraph.

Methods:
(i) Line 133 – There appears to be an issue with the insertion of a symbol here with f() being in a dashed, square box.

Results:
(i) Line 209 – What is meant by ‘slightly invariable’?

Discussion:
(i) Lines 272 & 356 –Why is the common name mentioned here and not the scientific name? (see earlier comment)

Conclusion:
(i) Lines 358-360 – I find this final sentence of the Conclusion a little confusing in what was otherwise a well-written and clear manuscript. I suggest clarifying this sentence to strengthen the ‘take-away’ message from this study.

Figures:
(i) Figure 2 (caption) – There is repetition of words in this caption that is not needed. I suggest shortening it to read “(a) Sex-specific length- and (b) age-frequency distributions of sampled Hipposcarus longiceps from Guam.”

Reviewer 2 ·

Basic reporting

The quality of the manuscript is so high, so I gave only a handful suggesting and comments.

Experimental design

The research question was well defined, and perfect for the journal.

Validity of the findings

Data is enough for this large sized species, and all result are well reliable.

Annotated reviews are not available for download in order to protect the identity of reviewers who chose to remain anonymous.

---

## Round 0.2 · accepted · Accept

The comments by the reviewers have been responded to well, and the paper is now acceptable for publication. I look forward to seeing the published version!